# Ultrarobust, tough and highly stretchable self-healing materials based on cartilage-inspired noncovalent assembly nanostructure

Yuyan Wang[1], Xin Huang[1] & Xinxing Zhang [1✉]

Self-healing materials integrated with excellent mechanical strength and simultaneously high healing efficiency would be of great use in many fields, however their fabrication has been proven extremely challenging. Here, inspired by biological cartilage, we present an ultra-robust self-healing material by incorporating high density noncovalent bonds at the interfaces between the dentritic tannic acid-modified tungsten disulfide nanosheets and polyurethane matrix to collectively produce a strong interfacial interaction. The resultant nanocomposite material with interwoven network shows excellent tensile strength (52.3 MPa), high toughness (282.7 MJ m$^{-3}$, which is 1.6 times higher than spider silk and 9.4 times higher than metallic aluminum), high stretchability (1020.8%) and excellent healing efficiency (80–100%), which overturns the previous understanding of traditional noncovalent bonding self-healing materials where high mechanical robustness and healing ability are mutually exclusive. Moreover, the interfacical supramolecular crosslinking structure enables the functional-healing ability of the resultant flexible smart actuation devices. This work opens an avenue toward the development of ultrarobust self-healing materials for various flexible functional devices.

[1] State Key Laboratory of Polymer Materials Engineering, Polymer Research Institute, Sichuan University, Chengdu, China. ✉email: xxzwwh@scu.edu.cn

Recently, highly strong, self-healable, and stretchable materials have been increasingly desirable for electronic skin[1–6], wearable electronic devices[7–12], and artificial muscles, etc.[13–19]. The healing of damage in polymers can be realized through the reversibility of dynamic bonds of cross-link polymer chains, which is able to extend the service life and improve reliability and durability of functional devices[20]. Nevertheless, self-healing materials prepared by single noncovalent bonds suffer from low stregth, especially for self-healing hydrogels and elastomers whose strength is usually <3.0 MPa (refs. [21–27]). Hence, it has been a long-standing challenge to acquire flexible composites integrated with high stretchability, outstanding mechanical strength, and high self-healing ability at the same time.

To date, the tensile strength of most self-healing materials is limited within 10.0 MPa (refs. [27–31]) due to the low strength nature of noncovalent bonds, and the weak interfacial interactions between fillers and polymer matrix. Introducing multiple dynamic bonds into polymeric materials have been proposed to deal with the trade-offs in fabricating high-performance self-healing materials. Aida et. al. reported a low-molecular weight polymer by multiple noncovalent cross-linking with tensile strength up to 26.5 MPa (ref. [32]). Xi Zhang and coworkers designed and fabricated both reversible noncovalent bonds and permanent covalent cross-links to improve mechanical stength of elastomer to 34.0 MPa (ref. [33]). For most self-healing materials, their mechanical strength is still relatively low and not capable of meeting the requirement of structural materials.

In nature, the cartilage tissue of animals meets the above proposed requirements of structural materials, which has high mechanical strength and a certain self-healing ability after damage. Human cartilage tissue is composed of collagen cells and intercellular substance (Fig. 1a). The side chains of proteoglycan molecules in cartilage matrix are connected with collagen fibers through hydrogen bonds to form a network structure, and a large number of collagen fibers are interwoven into a network to bear high force (Fig. 1b). Therefore, such hierarchical structure with strong supramolecular interaction makes cartilage mechanically strong and tough. Previously, a number of studies have combined woven fiber networks or hard plastic skeletons with soft matrix to achieve ultrahigh mechanical properties[34–37]. This undoubtedly proves the effectiveness of the cartilage-like structure strategy of assembling a rigid "skeleton" into a flexible matrix by strong adhesion. However, since it is a macroscopic assembly of rigid networks and soft matrix, it is difficult for aforementioned composites to achieve high stretchability.

Recently, ultrathin two-dimensional (2D) tungsten disulfide (WS2) nanomaterials have attracted widespread attention for their excellent physical and chemical properties. However, its high rigidity and poor interfacial interaction with elastomer matrix make it difficult to adequately play a role in flexible smart actuation devices. At present, there is a lack of suitable routes to assemble such rigid 2D materials into appropriate polymer substrates to prepare self-healing materials. The structure of organs and tissues of living systems in nature gives precious inspiration to the structural design of materials. Combining high-

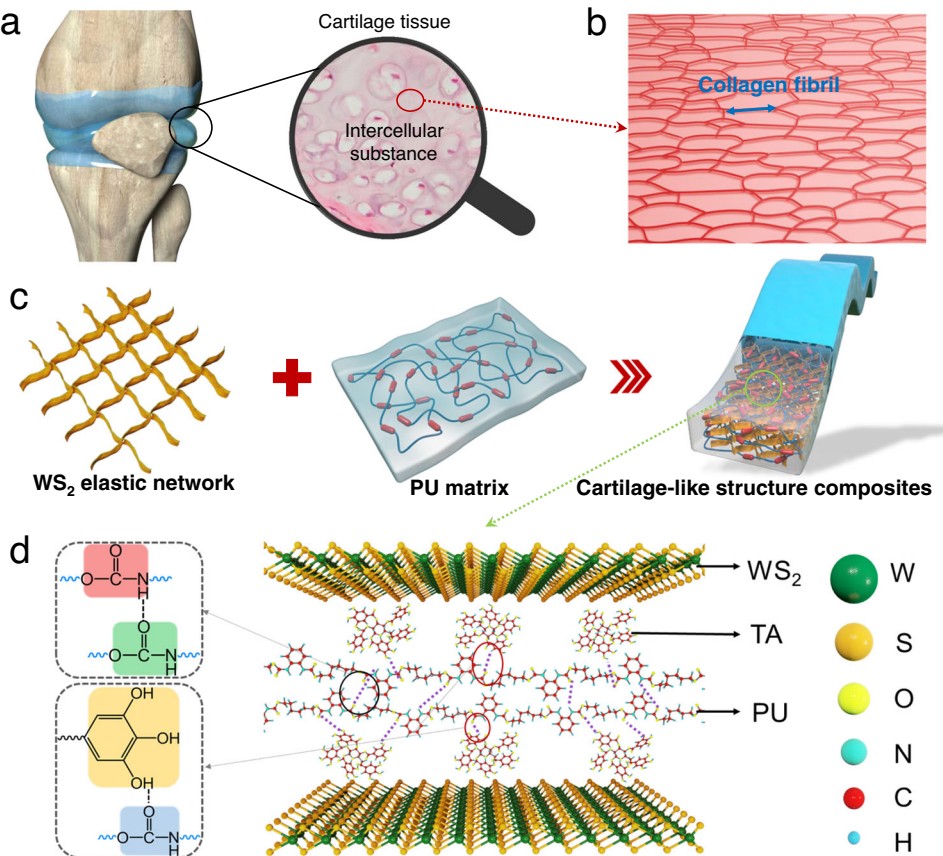

**Fig. 1 The nanostructure design of the cartilage-like PU composite. a** Schematic illustration of a cartilage structure. **b** Schematic illustration of intercellular substance of cartilage tissue. **c** Schematic illustration of the nanostructure of composite consisting of hydrogen-bonded interwoven network of 2D WS2 and PU matrix. **d** Schematics of the dynamic noncovalent bonding interaction between PU and interwoven network of 2D WS2.

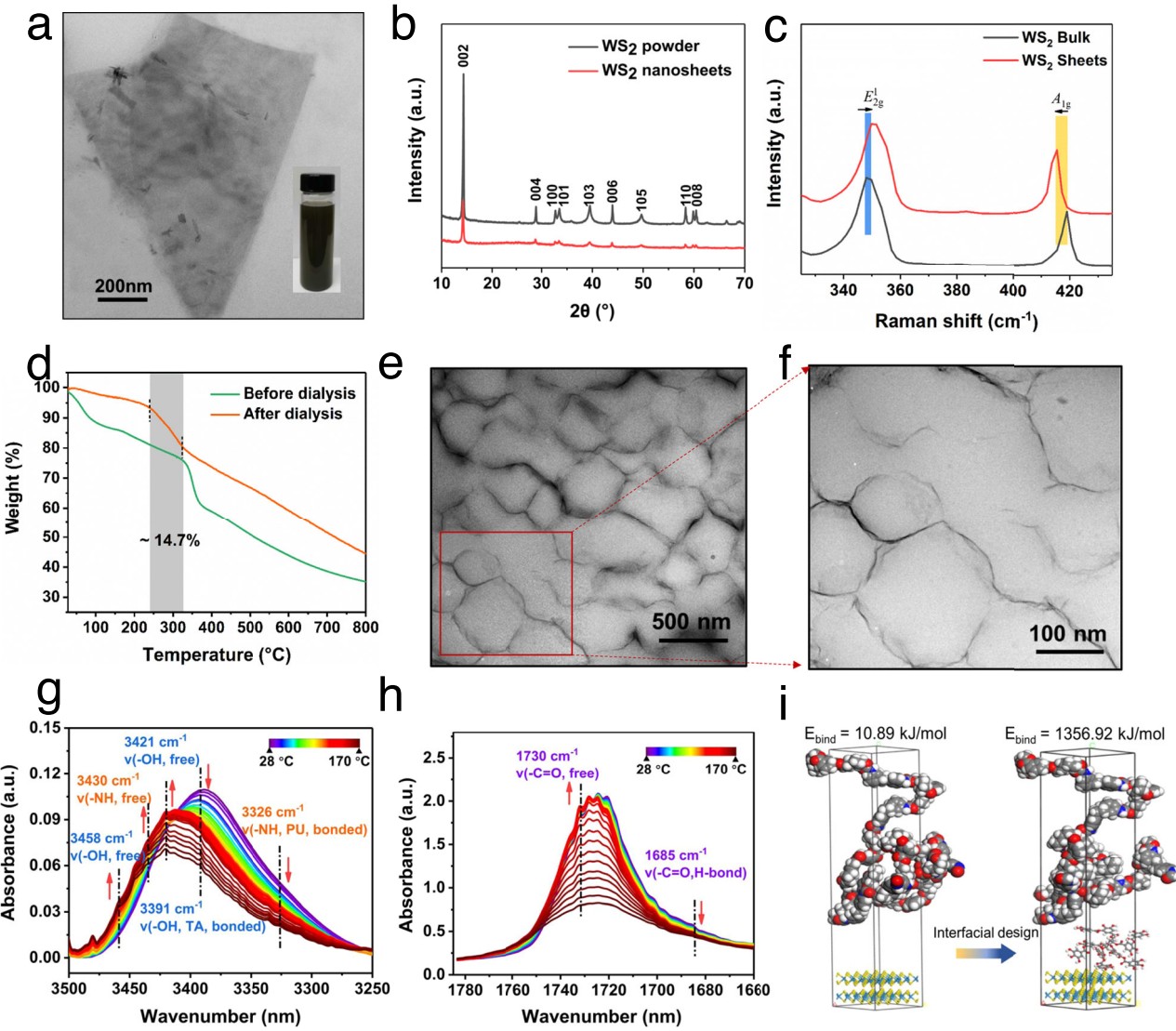

**Fig. 2 Structural and compositional characterizations. a** A high-resolution TEM image of the exfoliated WS$_2$ nanosheet. **b** XRD patterns of WS$_2$ powder and WS$_2$ nanosheets. **c** Raman spectra of WS$_2$ bulk and nanosheets. **d** TGA curve of TA-WS$_2$ nanosheets before and after dialysis. **e, f** TEM images of 16 wt% TA-WS$_2$/PU composites with 3D conductive network. **g, h** Temperature-dependent FTIR spectra of 16 wt% TA-WS$_2$/PU nanocomposites upon heating from 25 to 170 °C, **g** 3250–3500 cm$^{-1}$, and **h** 1660–1780 cm$^{-1}$. **i** Optimized structures and binding energies of the before and after interfacial design.

performance materials with rational bionic structure design is a versatile strategy to develop smart materials and flexible devices.

Here, we propose a cartilage-inspired microscale/nanoscale assembly route to fabricate ultrarobust self-healing materials based on noncovalent bonding-driven self-assembly of 2D nanosheets into an interwoven network. The interwoven network connected with dense hydrogen bonds aggregated at the interface endows the resultant composites with remarkably improved mechanical properties and self-healing efficiency, as well as excellent functional-healing ability of the developed flexible devices. Our results suggest an exciting material platform of high-performance self-healing materials, which could be applied to a wide range of flexible functional devices.

## Results

**Material design and characterization.** We embed a noncovalent bonding-driven self-assembled WS$_2$ network in waterborne polyurethane (PU) matrix to mimic the interwoven network structure of collagen fibrils in cartilage (Fig. 1c). Based on a tannin acid (TA)-assisted exfoliation strategy[38], we facilely

exfoliate WS$_2$ into monolayer or few-layer nanosheets through hydrophobic interaction between the 2D monolayer and hydrophobic aromatic structures in polyphenols. Then, we utilize the polyhydroxy structure of TA adsorbed on the nanosheets to construct and regulate the aggregation density of hydrogen bonds at the interface between 2D WS$_2$ nanosheets and PU matrix. This assembly process can retain the excellent physical and chemical properties of WS$_2$, which also has strong adhesion and lubricating characteristics during combination with PU matrix. Thus, the interwoven network of 2D WS$_2$ and the high-density hydrogen bonds at the interface synergistically endow the materials with high strength, toughness, and excellent self-healing abilities.

The morphology of as-prepared WS$_2$ nanosheets were evaluated by transmission electron microscopy (TEM), as shown in Fig. 2a. The a high-magnification TEM image unambiguously reveals the presence of a well-exfoliated WS$_2$ nanosheet. Particle size analysis of TA-modified WS$_2$ dispersion solution (Supplementary Fig. 1) shows that WS$_2$ nanosheets have an average size of 100 nm. The X-ray diffraction (XRD) pattern of the WS$_2$ nanosheets reveals highly crystalline the WS$_2$ phase with (002, 004, 100, 101, 103, 006,

105, 110, and 008) peaks identical to those of bulk $WS_2$ (JCPDS Card No. 08-0237) as shown in Fig. 2b (refs. [39,40]). The interlayer distance calculated according to Bragg's equation is 0.6 nm. From the Raman spectrum of $WS_2$ bulk (Fig. 2c), two characteristic bands located at 349.8 cm$^{-1}$ ($E^1_{2g}$) and 417.3 cm$^{-1}$ ($A_{1g}$) can be clearly distinguished. It is well known that the resonant Raman scattering of 2D materials is layer dependent[41,42]. After the exfoliation, the frequencies of these materials generally undergo small variations due to decreased interlayer interactions, i.e., the $E^1_{2g}$ band upshifts, while the $A_{1g}$ band downshifts, further indicating the presence of few-layer $WS_2$ nanosheets[43,44].

In addition, the unique dendritic polyhydroxy character of TA allows individual $WS_2$ nanosheets to be scaffolded to prevent aggregation. The absorbance change results in the UV–vis absorption spectra (Supplementary Fig. 2), indicating the as-exfoliated nanosheets can form highly stabilized dispersions without the appearance of aggregation phenomenon even after incubation for 30 days. The zeta potential of the TA-stabilized $WS_2$ nanosheets is −34.2 mV (Supplementary Fig. 3), which further indicates the stability of the nanosheets dispersion. After 1 week of dialysis, the obtained $WS_2$ nanosheets still attached ~14.7% TA (as shown in Fig. 2d)[45], as quantitatively measured by thermogravimetric analysis (TGA). This stems from the fascinating adhesion forces based on the catechol-inspired chemistry via multiple noncovalent interactions, and the origin of their interaction is mainly attributed to the hydrophobic interaction between the $WS_2$ monolayer and hydrophobic aromatic structures in polyphenols[46].

Based on the modification and stabilization of $WS_2$ nanosheet by TA, we assembled a cartilage-like structured nanocomposite. Due to the strong electronegativity of TA-$WS_2$ nanosheets, the nanosheets are directionally distributed around the negatively charged PU microspheres through electrostatic repulsion. Furthermore, the branches of TA possess multiple hydrogen-bonding sites, including multiple ester groups and phenolic hydroxyls. It is envisioned that such structure enables the TA molecule to entrap and bind PU chains through hydrogen bonds, maximizing the bonding chance between the hydrogen-bonding sites and easy to form high-density hydrogen bonds. Therefore, the strong hydrogen bond interaction induced the TA-$WS_2$ nanosheets to self-assemble into an interwoven network around the PU latex microsphere. The cross-section morphologies of TA-$WS_2$/PU nanocomposites (Fig. 2e, f) show that there is an ordered interwoven network of $WS_2$ nanosheets in PU matrix, like collagen fibrils in cartilage matrix. In contrast, we dried the mixture of $WS_2$ and PU latex directly after vacuum filtrating. As shown in Supplementary Fig. 4a, the resulting elastomer without hydrogen bonding interaction is clearly stratified due to the settled fillers in latex, as observed via scanning electron microscope. Also, the contrast sample has no satisfactory enhancement effect on mechanical properties (Supplementary Fig. 4b). The formation of hydrogen bonding between TA-$WS_2$ and PU was investigated by temperature-dependent Fourier transform infrared (FTIR) spectra, which has been proved a powerful tool to investigate the molecular interaction in polymers[47]. The temperature-dependent FTIR spectra of TA-$WS_2$/PU upon heating from room temperature to 170 °C are shown in Fig. 2g, h. For PU, the bands at 3391 cm$^{-1}$ (hydrogen-bonded –OH groups in TA, Supplementary Fig. 5) gradually shift to 3321 cm$^{-1}$. Besides, the intensity of 3326 cm$^{-1}$ related to hydrogen-bonded –NH groups in PU[48] gradually decreases, while the intensity of the bands at 3458 and 3430 cm$^{-1}$ gradually increases. These spectral features indicate that the hydrogen bonds between PU and TA-modified $WS_2$ are gradually broken with the increase of temperature. As for the –C=O groups of TA-$WS_2$/PU (Fig. 2f) composites, it is noted that the bands intensity

of the hydrogen-bonded –C=O group keeps decreasing in the whole heating process, but the peak intensity of free –C=O group ~1731 cm$^{-1}$ is increased[49,50]. These results indicate the breakage of various hydrogen bonds of related groups which generates "free" –C=O groups. Dynamic mechanical analysis (DMA) was further performed to illustrate the formation of supramolecular interactions in the PU matrix (Supplementary Fig. 6). The TA-$WS_2$/PU elastomer with supramolecular interactions has a higher glass transition temperature (15.75 °C) than that of the $WS_2$/PU elastomer (7.35 °C). TA-$WS_2$ nanosheets have large specific surface area and can provide abundant hydrogen bonds to form strong interactions with PU chains. As a result, these nanosheets have restraining effect on the polymer molecular chain movement to make the glass transition temperature move to higher temperature.

Binding energy simulation between PU molecular chain and the monolayer $WS_2$ nanosheet (TA modified or unmodified) was employed to further understand the interaction mechanism of noncovalent bonds (Fig. 2i). The binding energy can be calculated according to the following equation[51,52]:

$$E_{bind} = E_{PU} + E_{filler} - E_{total} \qquad (1)$$

where $E_{bind}$ is the binding energy between PU and filler; $E_{PU}$ and $E_{filler}$ represent the corresponding energy of PU and filler in the optimized conformation, and $E_{total}$ is the total energy of the system. In order to compute the binding energy between PU and filler, the model after molecular dynamic simulation of composite using the universal force field should be constructed to calculate $E_{total}$. The corresponding energy PU was calculated from the structure created from the optimized conformation of composite by removing the filler without further minimization, and the $E_{filler}$ was obtained by the same way after removing all the PU out from the optimized conformation of composite. Similarly, we can get the binding energy of polymer matrix and fillers from the optimized conformation of nanocomposites. It is worth noting that the binding energy between PU and TA-$WS_2$ reaches up to 1356.9 kJ mol$^{-1}$, whereas the binding energy of PU and monolayer $WS_2$ nanosheets is only 10.9 kJ mol$^{-1}$. As a result, the adsorption of TA onto the $WS_2$ nanosheets can supply strong binding sites for $WS_2$ and PU matrix, which is expected to endow the nanocomposites with fascinating self-healing property with high mechanical performance.

**Mechanical and self-healing performances.** For evaluating the practicability of the nanocomposites as functional devices, its mechanical and self-healing properties were studied in detail. Benefiting from the cartilage-like nanostructure, the nanocomposites exhibit excellent mechanical and self-healing performance. To demonstrate the mechanical deformation of the composite elastomer, small-angle X-ray scattering (SAXS) analysis of 16 wt% TA-$WS_2$/PU composite sample and control 16 wt% $WS_2$/PU sample at different strains was conducted. The 2D scattering patterns are shown in Fig. 3a. As the strain increased, the scattering signal changed from homo-dispersion to orientation to the equator direction, and the scattering ring become rhombic, indicating that the isotropic $WS_2$ network structure was oriented along the stress direction. The direction of $WS_2$ bearing load was consistent with the direction of external force, which could effectively bear the stress of the matrix in the tensile direction and improve the strength of the material[53,54]. In the control sample, the oriented behavior is much weaker (Supplementary Fig. 7). Due to the lack of TA adhesion, the thickness of the interfacial layer decreased from 2.34 to 0.27 nm (Supplementary Fig. 8), and the $WS_2$ which did not form an elastic network, so that PU molecular chains could not drive the

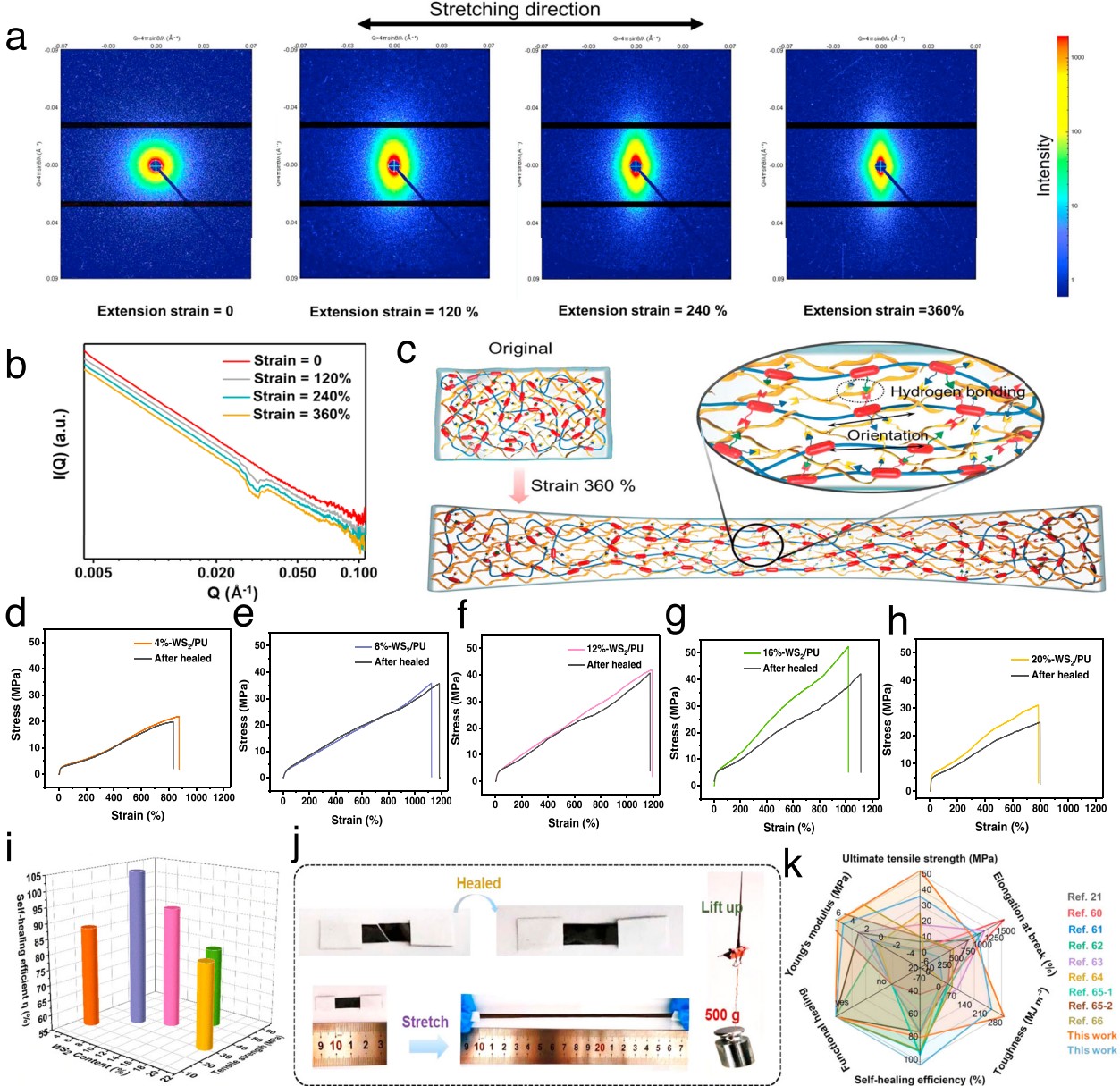

**Fig. 3 Mechanical and self-healing performances. a** 2D SAXS images of the 16 wt% TA-WS$_2$/PU with different tensile strain during uniaxial stretching process. **b** 1D scattering profiles of 16 wt% TA-WS$_2$/PU integrated from 2D SAXS patterns under different strains. **c** Schematic illustrations of the nanostructure of the original sample and stretching sample. **d–h** Mechanical self-healing performance of PU composites filled with different contents of TA-WS$_2$. **i** Comparison of tensile strength and self-healing efficiency of our TA-WS$_2$/PU nanocomposite with different filler content. **j** Photographs demonstrating excellent mechanical property of self-healed 16 wt% TA-WS$_2$/PU composites when stretched and lift a weight. **k** Comparison of Young's modulus, elongation at break, ultimate tensile strength, toughness, self-healing efficiency, and functional-healing ability of our nanocomposite with other self-healing materials.

orientation of WS$_2$. Figure 3b and Supplementary Fig. 9 show the representative scattering profiles of composite sample and control sample under different strains, respectively. An obvious increase in the scattering peak at $q \approx 0.032$ Å$^{-1}$ with increasing strain can be observed. The peak at $q \approx 1.3$ nm$^{-1}$ for composite with high strain can be ascribed to the break–reconstruction of hydrogen bonds that have lower electronic density in the stretch process, which contributes to the improvement of the mechanical strength compared with the control sample.

The reinforcing mechanism for the composites is illustrated in Fig. 3c. The dendritic polyhydroxyl TA molecule loaded on WS$_2$ nanosheets could act as physical cross-linking points by forming hydrogen bonding interactions in PU matrix. Furthermore, during the stretching process, the interfacial hydrogen bonds

between TA-WS$_2$ and PU facilitated the orientation of chain segments and led to strain-induced self-reinforcement. The physical cross-linking network and the self-reinforcement during stretching enhanced the strength of elastomer. Compared with traditional noncovalent bonding connections, the special contributions of TA-modified WS$_2$ include the formation of a microphase separation structure in the PU matrix and provide the interfacial dynamic hydrogen bonds. This structural construction could form a strong physical cross-linking network and restrain the PU chain segments to facilitate the strain-induced microphase separation, leading to a much stronger tensile strength and toughness of nanocomposites. As a result, the mechanical properties of our microscale/nanoscale-assembled cartilage-like nanocomposites (16 wt% TA-WS$_2$) are dramatically

improved, with the highest tensile strength of 52.3 MPa, toughness of 282.7 MJ m$^{-3}$, and elongation at break of 1020.8%, respectively, which are ~1.6, 8.0, and 16.4 times those of the pristine PU (Supplementary Fig. 10). Besides, the mechanical properties of the hybrid elastomer under different humidity (relative humidity (RH) 50, 60, 70, and 80%) were also investigated (Supplementary Fig. 11). The results indicate that humidity has little effect on the mechanical performance. Compared with other PU composites filled with rigid fillers (graphene, cellulose nanocrystalline, carbon nanotube, etc.), the as-prepared nanocomposite also shows outstanding mechanical performance (Supplementary Fig. 12), which further demonstrates the effectiveness of the controlled assembly of 2D materials and strong interfacial interaction design.

In addition, the dynamic fracture and reconstruction of hydrogen bonds dissipated energy continually, which improved the toughness of elastomer and enabled the nanocomposite to heal after damage. The mechanical and self-healing performances of TA-WS$_2$/PU composites with various filler content and hydrogen bonds density were measured by tensile tests, and the typical stress–strain curves are shown in Fig. 3d–h. After being cut into two pieces, these samples can recover their mechanical strength by bringing two freshly fractured surfaces into contact at room temperature and waiting for 12 h. The mechanical healing efficiency ($\eta$) is defined as the proportion of toughness restored relative to the original toughness, which takes into account the restoration of both strain and stress. As shown in Fig. 3i, the tensile strength and toughness of the measured TA-WS$_2$/PU composites increase and then decrease with the increase in TA-WS$_2$ contents of the TA-WS$_2$/PU composites. The 16 wt% TA-WS$_2$/PU composite shows the optimal mechanical performance with the self-healing efficiency of 80.6% due to the highest density of hydrogen bonds between the components. Also, the 8 wt% TA-WS$_2$/PU composite exhibits the highest self-healing efficiency (105.1%) without any external assistance. To investigate the mechanical self-healing ability, the cutting-off nanocomposite was reconnected for healing (Fig. 3j). The healed sample could stretch from 1.5 to 16.0 cm without break, and lift a weight of 500 g. Moreover, to better understand the influence of different contents of TA-WS$_2$ on our material system, differential scanning calorimetric (DSC) study was conducted (Supplementary Fig. 13). With the increase of the content of TA-WS$_2$, the glass transition temperature and melting temperature of the hybrid elastomer increase. It indicates that the increased content of TA-WS$_2$ nanosheets inhibits the thermal motion of PU molecular chains, and the self-healing property is dominated by the construction of abundant reversible dynamic hydrogen bonds. Hence, as the substitution amount of TA-WS$_2$ reaches 20%, the mechanical performance of the elastomer abruptly decreases, probably due to the aggregation of WS$_2$ in the PU matrix, and the excess WS$_2$ also hindered the movement of polymer chain segments and restricted the recovery of broken hydrogen bonds. Furthermore, repeated cyclic tensile tests were performed with a high strain of 400% for ten cycles (Supplementary Fig. 14). There is a larger hysteresis of the hybrid elastomer (16% TA-WS$_2$) than that of the pristine PU during the loading and unloading processes, as shown by the stress–strain curve. We define the efficiency of energy dissipation as the ratio between the integrated area in the hysteresis loop and that under the loading curve. And the efficiency of hybrid elastomer is remarkably improved because the noncovalent bonds break and dissipate the strain energy. These observations are similar to those of previously reported hydrogen-bonded cross-linked elastomers and hydrogels[55–59]. After healing for 12 h, the damaged sample recovered to its original state, indicating the excellent self-healing properties of this elastomer.

The mechanical and self-healing properties of the nanocomposites exceed those of most self-healing polymers (Fig. 3k, and details in Supplementary Table 1)[4,21,60–65], which makes them attractive for applications in flexible devices. Notably, the toughness of the 16%-WS$_2$/PU composite (282.7 MJ m$^{-3}$) outperforms most polymer structural materials. For instance, its toughness is 1.6 times higher than that of the ultrastrong dragline fiber of spider. Most remarkably, the toughness of the composite is much higher than that of plastic, like polyether–ether–ketone (16.3 MJ m$^{-3}$), polyamide (38.8 MJ m$^{-3}$), high-density polyethylene (151.2 MJ m$^{-3}$). Thanks to the reversible hydrogen-bonding cross-linked networks (Fig. 1d), our self-healing material has two distinctive advantages: (i) it exhibits relatively high mechanical ultimate strength and stretchability, which make it more robust than other elastomers for pratical applications. (ii) It can self-heal itself at room temperature with great self-healing efficiency and heal function. Thereby, the incorporation of nanofillers into polymer matrices via noncovalent bonding connection is an effective strategy to fabricate highly strong and tough nanocomposites, which provides a guidance to design self-healing structural materials.

**Robust NIR actuation and self-healing performances**. Some previously reported self-healing materials have achieved excellent healing capability, but many of them can only heal polymer matrix, functional filler network could not be repaired so that fine structure is easy to be damaged. As a demonstration, we fabricated near-infrared (NIR) actuators based on the mismatch of thermal expansion between two layers. Benefiting from the unique interconnected network design, the as-prepared material exhibits rapid photothermal conversion and heat transfer rate (Supplementary Fig. 15). The high photothermal conversion and heat transfer rate endow the nanostructured composites with excellent NIR actuating performance (Supplementary Fig. 16). As shown in Fig. 4a, it takes only 0.9 s to bend to maximum angle (137°) for the strip-shaped sample. The photoinduced mechanical force under NIR light irradiation was investigated as shown in Fig. 4b, under irradiation with NIR light, the actuator generates a large stress that increases with the light intensity, reaching ~6.9 kPa for the light power of 0.6 W. When turning off the NIR light, $t$ the stress drops significantly and then goes down to zero quickly. The cycle of force up and down with NIR light on and off, respectively, which can be repeated for five times without damping. Besides, the actuating properties of the nanocomposite can also be restored after self-healing at room temperature.

Interestingly, the self-healed (SH) sample has the same actuating speed and amplitude as the original sample (Fig. 4c), and generates similar actuation stress compared with the original sample (Fig. 4d). The cycle of force up and down can be repeated upon NIR light on and off, which demonstrates high reliability and stability of our actuators without apparent fatigue with fast NIR responsiveness. A biomimetic flower was fabricated which can undergo repeated closing and blooming in response to NIR (808 nm, 0.6 W) on and off (Fig. 4e and Supplementary Movie 1). The temperature variation during the blooming actuation could be observed in IR images (Supplementary Fig. 17). The crawling robot exhibits step-by-step crawling behavior upon NIR light on and off (Fig. 4f, Supplementary Movie 2, and details of force analysis in Supplementary Note 1). We believe the ultrarobust flexible devices with excellent self-healing ability has great potential in broad scientific and engineering fields.

**Discussion**
In summary, we have developed a cartilage-inspired ultrarobust self-healing material by incorporating high-density noncovalent

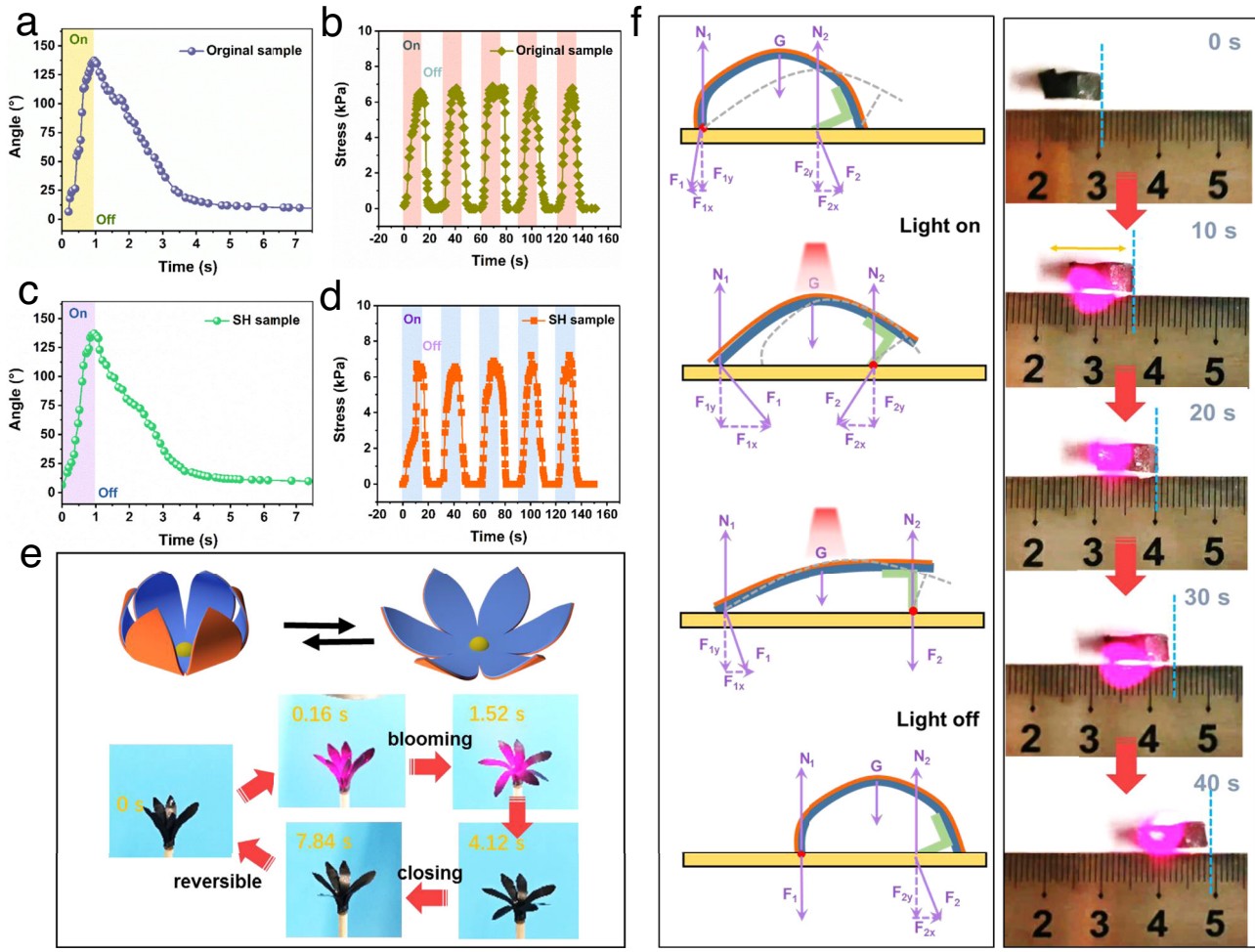

**Fig. 4 NIR light actuating and self-healing performances. a, c** Time dependence of the bending angle of **a** the original sample and **c** SH sample exposed to NIR light, when the light is switched on and off. **b, d** Actuation stress as a function of time as NIR light is periodically turned on (time, 15 s) and off (time, 15 s) on **b** the original and **d** SH sample. **e** Photographs of a "flower" blooming and closing stimulated by NIR light. **f** A crawling robot is able to continually crawl forward under periodic NIR light on and off.

bonds at the interface to collectively produce a strong adhesive force between assembled interwoven network of 2D TA-WS$_2$ nanosheets and PU matrix. The dynamic nature of the non-covalent bonds allows them to be broken and reformed during stretching, which leads to unfolding of the polymer chains and significantly increasing the thickness of the interfacial layer. In addition, the interwoven network of 2D TA-WS$_2$ nanosheets is similar to collagen nanofibrils in collagen matrix interwoven into a network to bear the force, which enables high stretchability, robustness, and self-healable ability of the material. We believe that the material's microscale/nanoscale structural design and high performance make it promising for artificial muscle applications, and envisage that the design concepts presented here may represent a general approach to the preparation of highly strong functional materials.

## Methods

**Materials**. WS$_2$, tannic acid (1.7 kDa, AR) was obtained from Shanghai Aladdin Biological Technology Co., LTD (China). Waterborne PU was purchased from Bayer Co., LTD (China). Cellulose nanofibers (CNF) 1% dispersion solution was purchased from Guilin Qihong Technology Co.,LTD (China). All the reagents were used as received without further purification. The water used in all experiments was deionized and ultrafiltered to 18.2 MΩ·cm with a Ulupure ultrapure water system.

**Polyphenol-assisted exfoliation of WS$_2$**. The exfoliation process was executed using an ultrasonic cell crusher with a tunable power from 0 to 1500 W. The tip of the ultrasonic cell crusher can utilize strong ultrasound to produce a cavitation effect in liquid to cause the solid particles or cellular tissue in liquid to break up, which is a more powerful ultrasonic method than that of a conventional bath sonication treatment. In brief, 240 mg of WS$_2$ powder was added into a 120 mL aqueous solution containing 120 mg of tannic acid, and then the suspensions were sonicated under 300 W for 2 h. Finally, the supernatant WS$_2$ nanosheets were collected after centrifugation at $4024.8 \times g$ for 15 min to remove unexfoliated WS$_2$. Note that ultrapure water (pH ≈ 6.0) was used during all the exfoliation experiments to prevent the aggregation of exfoliated nanosheets via oxidative cross-linking of polyphenols.

**Preparation of WS$_2$/PU nanocomposites**. Firstly, different amounts of TA-WS$_2$ suspensions were added into the PU latex suspension (1 g, 60 wt%) and were stirred for 30 min. Next, the TA-WS$_2$/PU film were fabricated via vacuum filtration of the mixed TA-WS$_2$/PU latex suspension on a PP filter membrane with 0.22 μm pore size to ensure the retained of the TA-WS$_2$/PU film on the filtration membrane. The resultant films were left in air at room temperature for 4 h to dry and then peeled off from the polypropylene membrane. The weight ratio of and TA-WS$_2$ and PU was 4%, 8%, 12%, 16%, and 20%, respectively.

**Preparation of actuators**. The CNF 1% dispersion solution was passed through the vacuum filtrated through a cellulose nitrate film to prepare freestanding CNF film. Once CNF film appeared to be dried, 10 mL TA-WS$_2$/PU latex was dispensed through it to form a bilayer film. The resulted nanocomposite film was left for air-drying at room temperature until it detached by itself from the cellulose nitrate membrane.

**Fabrication of bionic robots with reversible actuation**. The difference of thermal expansivity between the two sides of the film makes it possible to change shapes upon NIR light on and off. The patterned actuator was fabricated with the help of

origami or kirigami method. The reversible actuating performance was carried out by controlling the NIR light on and off with the help of commercial NIR laser device. As for the NIR response of the 2D film, the actuation angle of the film increases with the increase of NIR light power. Next, the actuation angle gradually decreases, and the sample return to the original 2D film shape with NIR light and turning off. The actuation movement of samples were obtained by a digital camera (D7100, Nikon), and the data of the actuation angles were captured from the camera record.

**Characterization**. TEM was performed on a transmission electron microscope (JEOL JEM-100CX, Japan). The XRD pattern was recorded from 10° to 70° using a Philips Analytical X'Pert X-diffractometer (The Netherlands) with Cu Kα radiation ($\kappa = 0.1540$ nm). Laser confocal microscopy Raman spectroscopy (HORIBA, HR Evolution, Japan) with a 532 nm laser line was performed to characterize the exfoliation of $WS_2$. The exfoliation concentration and yield of as-obtained $WS_2$ nanosheet dispersions were calculated by measuring the mass of $WS_2$ after freeze-drying the exfoliated dispersions (adsorbed TA on as-exfoliated nanosheet surface was subtracted from TGA analysis). FTIR spectroscopy was performed from 3500 to 700 cm$^{-1}$ on a Nicolet 6700 spectrophotometer (USA). DMA was performed on a TA Instrument (DMA Q800, USA) in strain-controlled mode. For temperature ramp experiments, rectangular film samples with dimensions of ~10 mm × 5 mm × 0.6 mm were run at a frequency of 1 Hz, with 10 μm amplitude and 125% force track ratio. Dynamic mechanical properties were measured from −50 °C up to 150 °C at 3 °C min$^{-1}$. Dynamic moduli were analyzed using TA Universal Analysis 2000. DSC curves were obtained using a TA Instrument (DSC Q1000, USA), equipped with a liquid nitrogen cooling system. Samples were run from −50 °C to 200 °C at a heating rate of 10 °C min$^{-1}$. The infrared imaging device (Fluke Ti400, USA) was performed to detect the temperature distribution of the samples.

**Mechanical and self-healing tests**. We characterized the mechanical properties of hybrid elastomers using Instron 5560 (USA) with a 1000 N load cell. Samples were cut from the film that had peeled off from the polypropylene membrane, using a normalized cutter. The samples have a central part of 15 mm in length, 2 mm in width, and 0.5–0.8 mm in thickness. Mechanical tensile stress and mechanical self-healing property tests were performed at room temperature with a stain rate of 100 mm min$^{-1}$. The temperature and RH at the time of the experiment were 24.2 °C and 46% RH, respectively. Damaged samples were given 12 h to heal at 25 °C and ~50% RH prior to tensile measurements. For mechanical property test under different humidity (RH 50, 60, 70, and 80%), samples were put in the constant temperature humidity chamber (Senya, GZX-350B) to balance humidity for 2 h and then perform the experiment. Values of the Young's modules, maximum strain at break, and healing efficiencies were determined according to data of at least three trials. Cyclic extension tests were performed on the same tensile machine cycling ten times with the maximum 400% strain at a strain rate of 100 mm min$^{-1}$.

**Bind energy simulation**. The molecular simulation is used to evaluate the interfacial design of the material system. It was performed under a universal force field in Material Studio 2018. First, models of TA molecule and the $WS_2$ (0 0 1) layer were constructed and geometrically optimized using the Forcite module. The $WS_2$ layer was modeled using 9 × 9 supercells. The vacuum slam which refers to the distance between $WS_2$ monolayers was set as 50 Å to avoid any nonbonding interactions between two adjacent layers. Then, in performing the adsorption locator simulation, the TA molecule was randomly distributed over the $WS_2$ layer with the max adsorption distance of 15 Å. Next, the amorphous structure of PU was constructed by a predigested method. Three molecules of PU with three degrees of polymerization were built and then geometrical optimization was performed. After each component was modeled, the overall system with initial energy minimization was subsequently built. To further obtain the optimized conformation, the isothermal–isochoric molecular dynamic simulation of composite model was conducted at 600 K. During the simulation, the positions of TA molecular and the $WS_2$ layer were fixed, but the PU chains were allowed to adjust conformation. Then, the binding energy can be calculated based on the final conformation model according to the following equation: $E_{bind} = E_{PU} + E_{filler} - E_{total}$ (1).

**Temperature-dependent FTIR spectroscopy**. A Nicolet iS50 Fourier transform spectrometer (U.S.A.) equipped with a deuterated triglycine sulfate detector was used for the temperature-dependent FTIR experiments. The TA-$WS_2$ with PU composite films were firstly sandwiched between two CaF$_2$ windows, and then placed into a home-made in situ pool (programmable heating device). The films were heated from 25 to 170 °C at 5 °C min$^{-1}$. Each FTIR spectrum was obtained from 20 scans with a resolution of 4 cm$^{-1}$, and totally 65 spectra was gathered during heating. During the experiment, the samples were protected by high-purity nitrogen gas (200 mL min$^{-1}$).

**In situ stretching SAXS measurement**. Scattering data were obtained from a SAXS apparatus Xeuss 2.0. The incident X-ray beam with an energy of 15 keV was used, and it was normal to the sample film so that the SAXS was in transmission geometry. A Rayonix MX225-HE CCD X-ray detector was used, with a sample-to-

detector distance of 3489.2 mm. The 2D CCD images were then reduced into scattering intensity ($I$) as a function of scattering vector ($q = 4\pi\sin\theta/\lambda$, where $\theta$ is half of the scattering angle and $\lambda$ is X-ray wavelength), using the software equipped at the beamline. In situ stretching SAXS measurements were based on a stretch station which can be controlled by wireless. SAXS data were recorded at different strain. For relaxation mode, the specimen was stretched to a strain of $\lambda = 4.5$, and then SAXS data were recorded during different relaxation time. The 2D SAXS scattering data were transformed into 1D SAXS curves by the default software of the SAXS apparatus.

**Photoinduced mechanical force testing**. As for photoinduced mechanical force, we carried out measurements at room temperature to monitor the light-generated force of TA-$WS_2$/PU film by applying NIR light (808 nm) to a strip-shaped sample (15 mm × 2 mm × 0.05 mm) held above an analytical balance, the photoinduced mechanical force generated in the sample sheet can be sensed and measured from the following formula:

$$F = mg \qquad (2)$$

where $m$ is a real-time indicator of the balance and $g$ is the acceleration of gravity.

## Data availability
The data that support the findings of this study are available from the corresponding author upon reasonable request.

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

## Acknowledgements

The authors thank the National Natural Science Foundation of China (51873123 and 51673121), the National Key Research and Development Program of China (2020YFC1909500), and the State Key Laboratory of Polymer Materials Engineering (Grant sklpme2019-2-20) for financial support. The authors also thank Dr. Guiping Yuan from the Analytical and Testing Centre of Sichuan University for providing the TEM measurement.

## Author contributions

X.Z. supervised the project. X.Z. and Y.W. conceived the project. Y.W. performed the experiments and wrote the paper. X.H. performed binding energy simulation. Y.W. performed the data analysis and wrote the paper. The other authors revised the manuscript and provided some suggestions.

## Competing interests

The authors declare no competing interests.
