## [Peer Review File · Nature Communications]

REVIEWER COMMENTS

Reviewer #1 (Remarks to the Author):

This interesting manuscript reported a bioinspired nanocomposite assembled by inorganic nanosheets and polyurethane matrix with robust mechanical strength and high self-healing efficiency. The method used to fabricate the materials with high performance is simple and effective. In addition, the mechanical properties and healing efficiency are impressive. However, the crucial characterizations of these materials are not sufficient. Overall, I believe this work has the potential to be appreciated by the readership of Nature Communications after major revisions. Some comments are listed below:

1. Fundamental thermal properties of the bioinspired nanocomposites should be fully characterized. The glass transition temperatures and melt temperatures can be obtained through DMA and DSC.
2. The details of self-healing experiments are missing. The operation procedures and experimental conditions should be provided.
3. The elasticity of the hybrid materials after adding the 2D nanosheets and the healing process should be carefully studied by cycling stretching tests.
4. The mechanical properties before and after the healing process are very important. Therefore, it is recommended that the typical tensile curves of the hybrid materials should be provided in Fig. 3.
5. The caption of Fig. 3j is "Comparison of Young's modulus, elongation at break, ultimate tensile strength, toughness, self-healing efficiency, and functional healing ability of our nanocomposite with other self-healing materials". However, the Young's modulus is missing. Therefore, the comparison of Young's modulus should be added in Fig. 3j.
6. There are some typo errors. For example: the vertical coordinate of Fig. S4 should be "Transmittance"; the "Fig. 4a" in the line 236 of page 13 should be "Fig. 1a". The authors should carefully check the manuscript and supporting information.

Reviewer #2 (Remarks to the Author):

This article presents a new self-healing elastomer nanocomposite based on 2D Tungsten disulphide nanosheets, functionalized with Tannic Acid, and a PU matrix. The results are of interest to the community, demonstrating a good combination of high toughness and elongation to break, but yet with a reasonably high tensile strength (50 MPa). Moreover, the material apparently is able to heal cuts, although precise details on this are lacking in the document. This combination of materials is novel, at least to the best of my knowledge, although inspiration from cartilage structure is not new in the field of novel bio-inspired materials, and deserves publication. However, the article in its present form lacks precision in the information and more insight in the mechanisms leading to these results. In particular, the role of the WS₂ content in the results is not analyzed (apart from 1 graph in the Supplementary materials, but would bring insight into the relative contribution of each physical parameters. More precise comments follow:

- First of all, the language should be revised, there are many grammatical errors, and strangely built sentences that make the reading tedious. (some examples, line 224, efficient instead of efficiency,

line 70: make cartilage mechanical strong, line 138: the verb is missing, line 176: the deformation mechanical of the composite...and there are many others).

- The abstract should indicate what materials have been used, instead of the vague: nanosheets and polymer matrix.

- in the intro, around lines 47-48 the schematic of the cartilage structure should already be presented, to better illustrate the work.

- Line 64 to 77 does not belong to the section on Results. Instead, this is background which should be in the introduction.

- line 86-87 and later...it is not clear at all why you should get an interwoven network of WS2 at this stage and later in the document. Certainly, this is a key aspect of the work and a main result of this work, as illustrated in the figure 2e and f; the WS2 structure assembles into a hexagonal pattern, which creates this network that may be crucial for the mechanical behavior. However, I do not see any discussion on this point. For me, looking into the experimental procedures, this is a direct consequence of the latex route used to process the samples, but there may be other physical effects linked to the processing route, drying method, etc. Or is the binding energy between PU and the WS-TA sufficient to explain the whole range of properties. Can the authors comment and evaluate the role of this structure? Have they tried different processing routes?

- in all results presented in the main document, there is no precision of the volume content of WS2 in the samples. Are these all the same composition? A range of compositions was investigated, as explained in the experimental section and in the supplementary information, what the samples are should be precisely indicated in the figures or the text.

- the experimental section is not complete and there is no mention of the measurement of mechanical properties, size of samples, machine used, how are the properties measured (in particular the modulus), and how many repeats of samples? How was the self-healing assessed (beyond a basic observation as reported in the main document showing repair by holding a weight (another typo in the text, by the way, it is weight not weigh)?). How many repeats, and what was the time left for healing, at what temperature?

- these materials are all very sensitive to humidity, due to the strong role of H bonds. Was the RH controlled in the sample storage, in the experiments? I could not find any mention to this in the document.

- line 337, what is a vacuum slam? Please explain, as the reader may not understand what this refers to.

-Table 1 in the supplementary section, please remove the % that appear in some of the boxes in the healing efficiency section.

Reviewer #3 (Remarks to the Author):

In short, mechanical properties of nanocomposite hydrogels are very impressive. Several minor comments should be addressed before publication.

1. First, writing needs to be improved. In the first several pages, particularly, in the abstract and conclusion, nothing is mentioned for this hydrogel system. It makes very difficult to understand what is the novelty of current study.

2. Simulation modeling in Figure 2i is poorly described, what methods are used to calculate the

binding energy? It is a common flaw that experiments applied an inaccurate modeling with many artifacts to quickly demonstrate the experimental observation.

3. what are water contents for different hydrogel systems? Water content is a critical parameter to control mechanical and actuation performance of hydrogels.

4. Where are the self-healing data? What are the tensile properties of self-healed hydrogels? Self-healing data should be presented in main text.

Dec. 09, 2020

Dear Editors,

Thank you very much for your efficient work in processing our manuscript entitled “Ultrarobust, tough and highly stretchable self-healing materials based on cartilage-inspired noncovalent assembly nanostructure” (Manuscript ID: NCOMMS-20-40345). We have carefully read the comments of the reviewers. Based on these comments and suggestions, we made careful modifications on the original manuscript through conducting a series of additional experiments. We feel that the revised manuscript is a great improvement on the original. All the revisions have been highlighted in blue color in the revised manuscript. We hope that the modified manuscript will meet the standard of the journal *Nature Communications*. Our point-by-point responses to the reviewers’ comments/suggestions are presented as follows:

To the reviewer 1:

This interesting manuscript reported a bioinspired nanocomposite assembled by inorganic nanosheets and polyurethane matrix with robust mechanical strength and high self-healing efficiency. The method used to fabricate the materials with high performance is simple and effective. In addition, the mechanical properties and healing efficiency are impressive. However, the crucial characterizations of these materials are not sufficient. Overall, I believe this work has the potential to be appreciated by the readership of *Nature Communications* after major revisions. Some comments are listed below:

We acknowledge your positive comments and suggestions. We have revised our manuscript in accordance with your instructive guidance. Hopefully we have addressed your concerns. Thank you very much.

- 1) Fundamental thermal properties of the bioinspired nanocomposites should be fully characterized. The glass transition temperatures and melt temperatures can be obtained through DMA and DSC.

√ The thermal properties of the bioinspired nanocomposites were characterized by DMA and DSC analyses according to the reviewer’s comments. The results were added in the Revised Supporting Information (Supplementary Fig. 6 and Supplementary Fig. 13), and relevant descriptions were provided in the Revised Manuscript (line 155-161, 253-259). Thank you for

your advice.

- 2) The details of self-healing experiments are missing. The operation procedures and experimental conditions should be provided.

✓ We have added the relevant operation procedures and experimental conditions of self-healing experiments in the Revised Manuscript (line 241-245, 389, 392) according to the reviewer's comments. Thank you for your pertinent advice.

- 3) The elasticity of the hybrid materials after adding the 2D nanosheets and the healing process should be carefully studied by cycling stretching tests.

✓ Based on the reviewer's concern, we evaluated the elasticity and healing process of the TA-WS₂/PU nanocomposites. The results were added in the Revised Supporting Information (Supplementary Fig. 14) and relevant descriptions can be found in Revised Manuscript (line 263-272). Thank you for your pertinent and helpful advice.

- 4) The mechanical properties before and after the healing process are very important. Therefore, it is recommended that the typical tensile curves of the hybrid materials should be provided in Fig. 3.

√ We have provided the typical tensile curves of the hybrid materials in Fig. 3 in the Revised Manuscript.

- 5) The caption of Fig. 3j is “Comparison of Young’s modulus, elongation at break, ultimate tensile strength, toughness, self-healing efficiency, and functional healing ability of our nanocomposite with other self-healing materials”. However, the Young’s modulus is missing. Therefore, the comparison of Young’s modulus should be added in Fig. 3j.

√ Based on the reviewer’s comments, we have added the Young’s modulus in Fig. 3k in the Revised Manuscript.

- 6) There are some typo errors. For example: the vertical coordinate of Fig. S4 should be “Transmittance”; the “Fig. 4a” in the line 236 of page 13 should be “Fig. 1a”. The authors should carefully check the manuscript and supporting information.

√ Based on the reviewer’s comments, we have checked the typos carefully in the Revised Manuscript and the Revised Supporting Information. Thank you very much for your suggestions.

To the reviewer 2:

This article presents a new self-healing elastomer nanocomposite based on 2D Tungsten disulphide nanosheets, functionalized with Tannic Acid, and a PU matrix. The results are of interest to the community, demonstrating a good combination of high toughness and elongation to break, but yet with a reasonably high tensile strength (50 MPa). Moreover, the material apparently is able to heal cuts, although precise details on this are lacking in the document. This combination of materials is novel, at least to the best of my knowledge, although inspiration from cartilage structure is not new in the field of novel bio-inspired materials, and deserves publication. However, the article in its present form lacks precision in the information and more insight in the mechanisms leading to these results. In particular, the role of the WS₂ content in the results is not analyzed (apart from 1 graph in the Supplementary materials, but would bring insight into the relative contribution of each physical parameters. More precise comments follow:

First of all, we acknowledge your comments and suggestions, which are valuable in improving the quality of our manuscript. We have added the relevant operation procedures

and details of self-healing experiments to Revised Manuscript. Furthermore, the influence of WS₂ content on the results and the assembly mechanism were explored. Hopefully we have addressed your concerns. We revised our manuscript in accordance with your instructive guidance and we feel that the revised manuscript is a significant improvement on the original one.

- 1) First of all, the language should be revised, there are many grammatical errors, and strangely built sentences that make the reading tedious. (some examples, line 224, efficient instead of efficiency, line 70: make cartilage mechanical strong, line 138: the verb is missing, line 176: the deformation mechanical of the composite...and there are many others).

√ Based on the reviewer's comments, we have checked and corrected the grammatical errors in the Revised Manuscript. Thank you very much for your suggestions.

- 2) The abstract should indicate what materials have been used, instead of the vague: nanosheets and polymer matrix.

√ Now the materials used have been clarified specifically in the abstract and conclusion according to the reviewer's comments. Thanks for your helpful suggestion.

- 3) in the intro, around lines 47-48 the schematic of the cartilage structure should already be presented, to better illustrate the work.

√ We have presented the cartilage structure properly in the introduction of the Revised Manuscript.

- 4) Line 64 to 77 does not belong to the section on Results. Instead, this is background which should be in the introduction.

√ The section about the structure and performance of cartilage has been moved to introduction in the Revised Manuscript. Thanks for your helpful suggestion.

- 5) line 86-87 and later...it is not clear at all why you should get an interwoven network of WS₂ at this stage and later in the document. Certainly, this is a key aspect of the work and a main result of this work, as illustrated in the figure 2e and f; the WS₂ structure assembles into a hexagonal pattern, which creates this network that may be crucial for the mechanical behavior. However, I do not see any discussion on this point. For me, looking into the

experimental procedures, this is a direct consequence of the latex route used to process the samples, but there may be other physical effects linked to the processing route, drying method, etc. Or is the binding energy between PU and the WS-TA sufficient to explain the whole range of properties. Can the authors comment and evaluate the role of this structure? Have they tried different processing routes?

√ For the WS₂ structure assemblies, due to the strong electronegativity of TA-WS₂ nanosheets, the nanosheets are directionally distributed around the negatively charged PU microspheres through electrostatic repulsion. Furthermore, the branches of TA possess multiple hydrogen-bonding sites including multiple ester groups and phenolic hydroxyls. It is envisioned that such structure enables the TA molecule to entrap and bind PU chains through hydrogen bonds, thus maximizing the bonding chance between the hydrogen-bonding sites and easy to form high density hydrogen bonds. Therefore, the strong hydrogen bonding interaction induced the TA-WS₂ nanosheets to self-assemble into an interwoven network around the PU latex microsphere. As a control experiment, we dried the mixture of WS₂ and PU latex directly after vacuum filtrating. As shown in Supplementary Fig. 4a, the resulting elastomer without hydrogen bonding interaction is clearly stratified due to the nanosheets settled in latex in scanning electron microscope (SEM) observation. Also, the control sample has no desirable enhancement effect on mechanical properties (Supplementary Fig. 4b). Thank you for your pertinent advice.

- 6) In all results presented in the main document, there is no precision of the volume content of WS₂ in the samples. Are these all the same composition? A range of compositions was investigated, as explained in the experimental section and in the supplementary information,

what the samples are should be precisely indicated in the figures or the text.

√ The effect of WS₂ content on the results and the assembly mechanism were explored in the Revised Manuscript (line 245-263). The results indicate the comprehensive performance of the composite materials can be optimized at 16 wt% contents of TA-WS₂. Therefore, other characterizations of TA-WS₂/PU composite materials were conducted based on this optimized content. According to the suggestions of reviewers, the contents of TA-WS₂ have been clearly marked in figures and text. Thank you for your pertinent advice.

- 7) The experimental section is not complete and there is no mention of the measurement of mechanical properties, size of samples, machine used, how are the properties measured (in particular the modulus), and how many repeats of samples? How was the self-healing assessed (beyond a basic observation as reported in the main document showing repair by holding a weight (another typo in the text, by the way, it is weight not weigh)?). How many repeats, and what was the time left for healing, at what temperature?

√ According to the reviewer's comments, we have improved the experimental section regarding the measurement of mechanical properties in the Revised Manuscript (line 386-397). We assessed the self-healing property through comparing the self-healing efficiency. We defined mechanical healing efficiency (η) as the proportion of toughness restored versus the original toughness, with consideration of the restoration of both strain and stress. Related test details (repetition times, healing time, healing temperature) have been added in the Revised Manuscript (line 241-245, 389, 392). And the typo mentioned has been corrected.

- 8) These materials are all very sensitive to humidity, due to the strong role of H bonds. Was the RH controlled in the sample storage, in the experiments? I could not find any mention to this in the document.

√ We conducted the whole experiment under constant temperature and humidity (temperature: 24. 2°C, humidity: 46%), which is shown in the Revised Manuscript (line 391). Moreover, no water was added during the self-healing and storage process.

- 9) line 337, what is a vacuum slam? Please explain, as the reader may not understand what this refers to.

√ The vacuum slam here refers to the distance between WS₂ monolayers. According to the reviewer's suggestion, the vacuum slam has been specified in the Revised Manuscript.

- 10) Table 1 in the supplementary section, please remove the % that appear in some of the boxes in the healing efficiency section.

√ According to the reviewer's comments, we have removed the % that appear in some of the boxes in the healing efficiency section. Thanks for your helpful suggestion.

To the reviewer 3:

In short, mechanical properties of nanocomposite hydrogels are very impressive. Several minor comments should be addressed before publication.

We acknowledge your positive comments and suggestions. We have revised our manuscript in accordance with your instructive guidance. Hopefully we have addressed your concerns. Thank you very much.

- 1) First, writing needs to be improved. In the first several pages, particularly, in the abstract and conclusion, nothing is mentioned for this hydrogel system. It makes very difficult to understand what is the novelty of current study.

√ We have improved the language in the Revised Manuscript. Based on the reviewer's suggestion, we have specified the materials system in the abstract and conclusion. Thanks for your helpful suggestion.

- 2) Simulation modeling in Figure 2i is poorly described, what methods are used to calculate the binding energy? It is a common flaw that experiments applied an inaccurate modeling with many artifacts to quickly demonstrate the experimental observation.

√ Based on the reviewer's suggestion, the details of simulation modeling have been supplemented in methods section of the Revised Manuscript. Specifically, according to the previous studies (*ACS Nano* **12**, 12347–12356 (2018); *Small* **6**, 205–209 (2010); *Polymer (Guildf)*. **43**, 5197–5207 (2002)), the technique of molecular simulation can be applied to evaluate the interactions among PU, TA and WS₂. Firstly, the models of PU, TA, WS₂ and their composite were constructed using different tools in Material Studio 2018. All structures were optimized using the Universal force field in Forcite module. Next, the molecular dynamic simulation of composite model was conducted to obtain the optimized conformation.

Then, the binding energy can be calculated based on the final conformation model according to the following equation: $E_{\text{bind}}=E_{\text{PU}}+E_{\text{filler}}-E_{\text{total}}$, where E_{bind} is the binding energy between PU and filler; E_{PU} and E_{filler} represent the corresponding energy of PU and filler, and E_{total} is the total energy of the overall system.

- 3) what are water contents for different hydrogel systems? Water content is a critical parameter to control mechanical and actuation performance of hydrogels.

√ Since our material system is based on a rubbery elastomer, it is not sensitive to water. According to the reviewer's comments, we performed the mechanical tests under different humidity (RH 50%, 60%, 70%, 80%). The results are shown in the Revised Manuscript (line 229-232, Supplementary Fig. 11). The results illustrated that water content has little effect on the mechanical performance. Similarly, there will be no change in actuation performance. Thank you for your pertinent advice.

- 4) Where are the self-healing data? What are the tensile properties of self-healed hydrogels? Self-healing data should be presented in main text.

√ Based on the reviewer's comments, we have provided the typical tensile curves of the hybrid materials and self-healing samples in Fig. 3 in the Revised Manuscript. Thank you very much for your suggestions.

Thank you very much for your assistance in this review process. The manuscript has been resubmitted to your journal. We look forward to your response. If there is something else needed, please do not hesitate to contact me directly by email listed below, and I will answer you as soon

as possible.

With best regards,

Sincerely yours,

Prof. Xinxing Zhang

State Key Laboratory of Polymer Materials Engineering

Polymer Research Institute, Sichuan University

Chengdu 610065, China

Tel: 86-28-85460607

Fax: 86-28-85402465

E-mail: xxzwwh@scu.edu.cn

REVIEWERS' COMMENTS

Reviewer #1 (Remarks to the Author):

The authors have adequately addressed all of the concerns. The manuscript can be accepted in the present form.

Reviewer #2 (Remarks to the Author):

The revised version addresses very well the main queries raised by the reviewers, the missing data or explanations are now added and the manuscript is fine. However, quite a few typos remain, to be checked, here are some: Page 2, line 40, typo: tentile (should be tensile), line 42: their strength are (should be "is"), line 262 page 15, , the recovery of break bonds (should be broken), line 270: hydrogen boned (should be bonded).

Reviewer #3 (Remarks to the Author):

the authors have addressed my concerns.

Dec. 30, 2020

Dear Editors,

Thank you very much for your efficient work in processing our manuscript entitled “Ultrarobust, tough and highly stretchable self-healing materials based on cartilage-inspired noncovalent assembly nanostructure” (Manuscript ID: NCOMMS-20-40345A). We have carefully read the comments of the reviewers. Based on these comments and editorial requests in the attached document, we made careful modifications on the original manuscript. All the revisions have been highlighted in blue color in the revised manuscript. We hope that the modified manuscript will meet the standard of the journal *Nature Communications*. Our point-by-point responses to the reviewers’ comments/suggestions are presented as follows:

To the reviewer 1:

The authors have adequately addressed all of the concerns. The manuscript can be accepted in the present form.

√ Thank you.

To the reviewer 2:

The revised version addresses very well the main queries raised by the reviewers, the missing data or explanations are now added and the manuscript is fine. However, quite a few typos remain, to be checked, here are some: Page 2, line 40, typo: tentile (should be tensile), line 42: their strength are (should be "is"), line 262 page 15, the recovery of break bonds (should be broken), line 270: hydrogen boned (should be bonded).

√ Thanks for your helpful suggestion. And the typo mentioned has been corrected.

To the reviewer 3:

the authors have addressed my concerns.

√ Thank you.

Thank you very much for your assistance in this review process. The manuscript has been resubmitted to your journal. We look forward to your response. If there is something else needed, please do not hesitate to contact me directly by email listed below, and I will answer you as soon as possible.

With best regards,

Sincerely yours,

Prof. Xinxing Zhang

State Key Laboratory of Polymer Materials Engineering

Polymer Research Institute, Sichuan University

Chengdu 610065, China

Tel: 86-28-85460607

Fax: 86-28-85402465

E-mail: xxzwwh@scu.edu.cn